# *Rubia akane* Nakai Fruit Extract Improves Obesity and Insulin Sensitivity in 3T3-L1 Adipocytes and High-Fat Diet-Induced Obese Mice

**DOI:** 10.3390/ijms26051833

**Published:** 2025-02-20

**Authors:** Juhye Park, Eunbi Lee, Ju-Ock Nam

**Affiliations:** 1School of Food Science and Biotechnology, College of Agriculture and Life Sciences, Kyungpook National University, Dague 41566, Republic of Korea; pdw8609@knu.ac.kr (J.P.); 21eunbi@knu.ac.kr (E.L.); 2Research Institute of Tailored Food Technology, Kyungpook National University, Dague 41566, Republic of Korea

**Keywords:** *Rubia akane* Nakai fruit, anti-obesity, adipogenic differentiation, lipogenesis, insulin sensitivity, AMPK

## Abstract

A rise in obesity during the COVID-19 pandemic has spurred the development of safe and effective natural anti-obesity agents. In this study, we propose *Rubia akane* Nakai fruit extract (RFE) as a potential natural product-based anti-obesity agent. *R. akane* Nakai is a plant of the Rubiaceae family that grows throughout Republic of Korea. Its roots have long been used medicinally and are known for various bioactivities, but the fruit’s bioactivities are unexplored. We investigated the anti-obesity effects of RFE using 3T3-L1 adipocytes and high-fat diet-induced obese mice. In 3T3-L1 adipocytes, RFE inhibited adipogenic differentiation and lipogenesis by downregulating PPARγ (peroxisome proliferator-activated receptor γ), C/EBPα (CCAAT enhancer-binding protein α), and SREBP-1 (sterol regulatory element-binding protein 1) through AMPK (AMP-activated protein kinase) activation and by delaying the initiation of MCE (mitotic clonal expansion), which is essential for early adipogenesis. At the in vivo level, RFE improved the phenotypes of obesity and insulin resistance. In white adipose tissue, RFE not only suppressed adipogenic differentiation and lipogenesis through AMPK activation but also improved insulin sensitivity by upregulating basal GLUT4 (glucose transporter type 4) expression. Therefore, this study advances RFE as a potential natural treatment for obesity and insulin resistance.

## 1. Introduction

The COVID-19 pandemic has increased social isolation, reducing physical activity and negatively impacting eating habits, which has contributed to a secondary obesity epidemic. Obesity rates have surged, with prevalence among men and women increasing from 11% and 15% to 25.3% and 42.4%, respectively [1]. The World Health Organization (WHO) expects that by 2025, 167 million adults and children will face health complications related to being overweight or obese [2].

Obesity, characterized by excessive lipid accumulation, can increase the risk of developing complications such as type 2 diabetes and cardiovascular diseases. In particular, ectopic lipid accumulation due to obesity induces lipotoxicity, which elevates the secretion of free fatty acids and reactive oxygen species (ROS), impairing insulin signaling and sensitivity. This insulin resistance disrupts insulin’s action in insulin-target tissues, such as adipose tissue, thereby hindering glucose transport into cells [3]. This in turn elevates overall blood glucose levels, becoming a major risk factor for type 2 diabetes.

Anti-obesity medications widely used worldwide fall into three categories: appetite suppressants, fat absorption inhibitors, and glucagon-like peptide 1 (GLP-1) analogues. However, each of these medications is associated with a range of side effects. Appetite suppressants, such as phentermine and amphetamine, exert their effects through the central nervous system and may lead to cardiovascular complications, including hypertension and cardiac arrhythmias. Orlistat, a selective inhibitor of pancreatic lipase, can also lead to gastrointestinal side effects, including abdominal distension and fatty stools. GLP-1 analogues, such as liraglutide (Saxenda), are widely used to treat obesity and diabetes by regulating insulin secretion and food intake, but they have recently been found to increase the risk of acute pancreatitis [4,5].

Due to the side effects of anti-obesity medications, there is a growing interest in relatively safe, natural product-based alternatives. Various compounds derived from natural products have been shown to possess anti-obesity effects [6], and are being proposed as alternatives for the treatment of obesity. In this study, we propose *Rubia akane* Nakai fruit extract (RFE) as a potential candidate as a natural product-based anti-obesity treatment. *R. akane* Nakai, belonging to the Rubiaceae family, grows wild throughout Republic of Korea and has been a key medicinal resource in traditional Eastern medicine and folk remedies for centuries. In previous studies, the genus *Rubia* has been reported to exhibit anticancer, anti-inflammatory, antioxidant, antibacterial, antidiabetic, anti-arthritic, anticonvulsant, and antispasmodic effects [7]. The roots and rhizomes of *Rubia* are rich in compounds such as quinones, terpenes, and cyclopeptides, which are considered the primary bioactive components of *Rubia*. Among them, quinone compounds such as mollugin, alizarin, 1-hydroxy-2-methyl-9,10-anthraquinone, rubiadin, and rubiquinone are primarily involved in the bioactivities of *Rubia* [8,9]. Therefore, most researchers have focused on the bioactivity and metabolites of Rubiae radix (the root of *R. akane* Nakai). In particular, Rubiae radix has been shown to exhibit bioactivities, including anti-obesity effects [10,11]. However, studies on the bioactivities of the fruit remain nonexistent. Previous studies have confirmed that the deep-purple and black fruits of *Rubia* are rich in anthocyanins, including cyanidin-3-O-glucoside, delphinidin-3-O-rutinoside, and cyanidin-3-O-rutinoside [12,13]. Additionally, the fruits are abundant in natural flavonoids, which are major polyphenolic compounds commonly found in a variety of vegetables and fruits. Among them, rutin, known to exhibit anti-obesity activity in 3T3-L1 adipocytes, has been identified, and notably, it is absent in the roots [14,15]. From the perspective of resource utilization, the use of only roots has limitations, as it excludes the aerial parts of the plant, while fruits offer the advantage of being a sustainably and stably sourced material.

This study characterizes the inhibitory effects of RFE on adipogenic differentiation and lipogenesis and identifies the activation of AMPK as the mechanism of its action in 3T3-L1 adipocytes and high-fat diet-induced obese mice. Additionally, it confirms an improvement in insulin sensitivity by demonstrating that RFE regulates the basal expression levels of GLUT4. Also, rutin is ascertained to be a bioactive component of RFE that may partially contribute to these effects using liquid chromatography–mass spectrometry (LC-MS) and high-performance liquid chromatography (HPLC) analyses. Based on these findings, RFE is proposed as a promising natural candidate for the prevention and treatment of obesity and insulin resistance.

## 2. Results

### 2.1. RFE Inhibits Adipogenic Differentiation and Lipogenesis in 3T3-L1 Adipocytes

In this study, we assessed the inhibitory effects of RFE on adipogenic differentiation and lipogenesis in 3T3-L1 adipocytes. First, we evaluated cytotoxicity to find the optimal, non-toxic RFE dose for adipocytes. The 3T3-L1 preadipocytes showed no cytotoxicity upon RFE treatment at doses up to 25 μg/mL for 24 h (Appendix A). Therefore, we ascertained the anti-obesity effects of RFE at doses of 12.5 μg/mL and 25 μg/mL.

During adipocyte differentiation, RFE treatment decreased oil red O content, indicating that it could inhibit lipid accumulation in adipocytes (Figure 1A,B). A cascade of key transcription factors modulates adipocyte differentiation. Positive regulators like PPARγ and C/EBPα are upregulated, while negative regulators such as GATA2 (GATA-binding protein 2) and CHOP10 (DDIT3: DNA damage-inducible transcript 3) are downregulated, causing differentiation into mature adipocytes [16]. Therefore, we investigated RFE-induced alterations in the mRNA expression of key transcription factors and found that RFE significantly suppressed the expression of *Pparγ* and *C/ebpα* while increasing the expression of *Gata2* and *Chop10* (Figure 1C). This led to a reduction in lipid accumulation markers expressed during the late stage of differentiation (Figure 1D). Additionally, RFE significantly suppressed the protein levels of key transcription factors (PPARγ and C/EBPα) and the lipid biosynthesis factor SREBP-1. In contrast, AMPK activation was significantly increased by RFE at a dose of 25 μg/mL (Figure 1E,F). The activation of AMPK in adipocytes can produce anti-obesity effects by inhibiting adipocyte differentiation and lipogenesis [17]. As such, we believe that RFE-induced AMPK activation contributed to the suppression of PPARγ, C/EBPα, and SREBP-1. Therefore, our results suggest that the anti-obesity effects of RFE are mediated by the inhibition of adipogenic differentiation and lipogenesis through AMPK activation in 3T3-L1 adipocytes.

### 2.2. RFE Inhibits Adipogenesis in the Early Phase of Differentiation

This study demonstrates that RFE strongly regulates the early phase of differentiation. Adipocyte differentiation occurs in three distinct phases: early, intermediate, and late. During early differentiation, preadipocytes that have stopped growing due to contact inhibition resume the cell cycle by MCE (mitotic clonal expansion), a process crucial for early adipogenesis. Subsequently, in the intermediate and late phases, the formation and maturation of lipid droplets occur through the expression of adipogenesis-related genes. Therefore, delaying the initiation of MCE in the early phase of differentiation may serve as a strategy to inhibit adipogenesis [18]

We ascertained that RFE inhibits adipogenic differentiation and lipogenesis (Figure 1). Based on this, we investigated the effect of RFE in each differentiation phase by applying RFE during differentiation according to the schedule described in Figure 2A [19]. Groups C, D, and H, which were treated in the early phase (days 0–2), with over 50% less oil red O content than the groups treated at the middle and late stages (Figure 2B,C). This trend was also observed in the mRNA expression levels of *Pparγ* and *C/ebpα*, which are key regulators of adipogenesis (Figure 2D,E). This indicates that RFE strongly regulates the early phase of adipogenesis.

We also ascertained the effect of RFE on cell cycle regulation during early differentiation using flow cytometry. Approximately 14 h after induction of differentiation, progression from the G0/G1 phase to the S phase begins, with peak DNA replication occurring around 18 h [18]. So, we examined the changes in the cell cycle at 16 h and 24 h after the initiation of MCE. We found that RFE suppressed the progression of the cell cycle into the S phase and maintained high relative G0/G1-phase cell levels at all time points compared to the control group (WC) (Figure 2F). Therefore, our results suggest that RFE inhibits initiation of MCE, delaying cell cycle resumption and ultimately preventing hyperplasia of adipocytes.

### 2.3. RFE Improves High-Fat Diet-Induced Obesity and Dyslipidemia

This study demonstrates the effects of RFE in improving obesity and dyslipidemia induced by a high-fat diet (HFD) in vivo. In the above subsections, we ascertained the anti-obesity effect of RFE at the in vitro level (Figure 1 and Figure 2). We confirmed the anti-obesity effect of RFE in vivo through an HFD-induced obese mouse model, as shown in Figure 3A. Administration of RFE significantly mitigated weight gain induced by an HFD (Figure 3B,C). This reduction was not influenced by food intake (Appendix A). Also, RFE significantly lowered the size and weight of white adipose tissue (Figure 3D,E). This led to a decrease in adiposity, which represents the percentage of body fat (Figure 3F). However, there was no change in the weight of brown adipose tissue or other metabolic organs (Appendix A).

Plasma cholesterol (CHO) and triglyceride (TG) levels, indicators of obesity-induced dyslipidemia, also significantly decreased in the high-dose RFE group (Figure 3G,H). Elevated high-density lipoprotein cholesterol (HDL-C) levels in the HFD group may be a compensatory response to increased blood TG levels [20]. Recently, the TG/HDL-C ratio has been suggested as a more effective indicator of plasma lipid status than individual low-density lipoprotein cholesterol (LDL-C), HDL-C, and TG levels [21], and RFE also reduced the TG/HDL-C ratio (Figure 3I). Additionally, through plasma biomarkers of hepatotoxicity and nephrotoxicity, we confirmed that RFE did not cause in vivo toxicity (Appendix A). Therefore, our results suggest that RFE can effectively alleviate obesity and dyslipidemia induced by an HFD in vivo.

### 2.4. RFE Inhibits Adipogenic Differentiation and Lipogenesis in White Adipose Tissue

This study demonstrates that RFE exerts its anti-obesity effect through AMPK activation in white adipose tissue. In Section 2.3, we demonstrated that RFE can mitigate high-fat diet-induced obesity. We next inspected the molecular mechanisms by which RFE influences adipogenic differentiation and lipogenesis in adipose tissue. First, we confirmed that the RFE treatments reduced lipid droplet sizes by H&E staining of white adipose tissue, suggesting that RFE could improve hypertrophy caused by lipid accumulation in adipose tissue (Figure 4A,B). Then, we gauged the protein expression levels of adipogenic and lipogenesis-related factors in white adipose tissue. We found that RFE significantly suppressed the expression levels of PPARγ, C/EBPα, and SREBP-1 while significantly activating AMPK signaling (Figure 4C–F), which was similar to the results in vitro (Figure 1F,G). The activation of AMPK in adipose tissue is involved in overall lipid metabolism through the downregulation of PPARγ, C/EBPα, and SREBP-1 [17]. Therefore, our results suggest that RFE can inhibit adipogenic differentiation and lipogenesis by activating AMPK in white adipose tissue, ultimately improving adipose tissue hypertrophy.

### 2.5. RFE Improves Systematic Insulin Sensitivity in Obese Mice and 3T3-L1 Adipocytes

This study demonstrates that RFE improves obesity-induced insulin sensitivity by regulating basal GLUT4 expression. Insulin, which is secreted by pancreatic β cells, regulates glucose and lipid homeostasis in insulin target tissues such as adipose tissue, liver, and skeletal muscle. Decreased insulin sensitivity impairs glucose transport into cells, causing higher blood glucose levels and increased insulin secretion through compensatory mechanisms. Ectopic lipid accumulation resulting from obesity significantly diminishes insulin sensitivity, with over 80% of obese patients experiencing insulin resistance at some stage [22]. So, we investigated whether RFE’s anti-obesity effect could also improve insulin resistance.

We first found that RFE significantly reduced fasting blood glucose and fasting insulin levels in obese mice (Figure 5A,B). This led to a decrease in HOMA-IR levels, an indicator of insulin resistance (Figure 5C). To further inspect the effect of RFE on insulin sensitivity, IP-GTTs (intraperitoneal glucose tolerance tests) were executed. The RFE group consistently exhibited blood glucose levels lower than those of the HFD group at all measured time points. Additionally, blood glucose levels in the high-dose RFE group tended to decrease after 30 min and returned to a level similar to baseline (0 min) level at 120 min (Figure 5D,E). This indicates that RFE can improve obesity-induced glucose intolerance and insulin sensitivity.

Also, we investigated whether RFE could regulate GLUT4 and found that RFE enhanced basal GLUT4 protein levels in white adipose tissue (Figure 5F). This trend was also observed in 3T3-L1 adipocytes (Figure 5G). Therefore, our results suggest that RFE improves systematic insulin sensitivity by upregulating basal GLUT4 expression in adipocytes and white adipose tissues.

### 2.6. Rutin Is a Bioactive Component in RFE That Has Anti-Obesity Effects and Improves Insulin Sensitivity

To ascertain the bioactive components in RFE that exhibited anti-obesity effects and improved insulin sensitivity, we performed LC-MS and HPLC analyses. We first focused on rutin, a flavonoid compound reported in plants of the genus *Rubia*. Rutin is known to activate AMPK in 3T3-L1 adipocytes, inhibiting lipid accumulation, and promoting GLUT4 expression to enhance glucose uptake [14,23]. This mechanism is similar to the anti-obesity effects observed with RFE in 3T3-L1 adipocytes. To confirm the presence of rutin in the RFE used in this study, we performed LC-MS analysis to profile the compounds in RFE. A peak corresponding to a molecular weight of 610.5 m/z, presumed to be rutin, was detected in the positive ion-mode LC-MS chromatogram (Figure 6A). Based on this, we further confirmed through HPLC analysis that RFE exhibited a peak at the same retention time (12.563 min) as rutin (Figure 6B). This suggests that rutin is contained in the RFE used in this study. Additionally, we confirmed that RFE contains rutin at a concentration of 1.085 mg/mL through a standard curve based on different concentrations of rutin (Appendix A). Therefore, we propose that rutin is a bioactive component of RFE that inhibits adipogenic differentiation and lipogenesis while enhancing insulin sensitivity in obesity.

## 3. Discussion

This is the first study to ascertain the anti-obesity effects and the improvement in insulin resistance associated with RFE. We confirmed that RFE can inhibit adipogenic differentiation and lipogenesis by activating AMPK in adipocytes and delaying the initiation of MCE during early differentiation. In addition, RFE improved HFD-induced obesity and dyslipidemia in obese mice. It significantly inhibited adipogenic differentiation and lipogenesis via the activation of AMPK in white adipose tissue and also improved glucose intolerance and insulin resistance by upregulating basal GLUT4 expression.

Obesity is defined by an excessive expansion of adipose tissue mass, which occurs due to an increase in the formation of new adipocytes (hyperplasia) or the enlargement of existing adipocytes (hypertrophy). Adipocyte hyperplasia is accompanied by adipogenesis, which corresponds to the differentiation of preadipocytes into mature adipocytes. Adipogenesis is controlled by a cascade of transcription factors. PPARγ and C/EBPα are key transcription factors for adipogenesis. A nuclear hormone receptor, PPARγ is essential for maintaining the phenotype of mature adipocytes [24]. Similarly, C/EBPα has been extensively surveyed for its role in regulating adipocyte differentiation and maintenance [25]. They act cooperatively to induce the expression of lipid accumulation factors, such as FAS, LPL, Ap2, and adiponectin, in the late stages of adipogenesis [26]. Additionally, undifferentiated preadipocytes express negative regulators of CHOP, GATA, and Wnt/β-catenin signaling at high levels. These negative regulators are essential for maintaining the undifferentiated state [16], and are suppressed upon differentiation. Thus, adipogenesis can be controlled through the modulation of transcription factor expression, which is proposed as a promising strategy for the treatment and prevention of obesity [27]. In this study, we ascertained that RFE suppressed the expression levels of PPARγ and C/EBPα in adipocytes while maintaining the expression of negative regulators CHOP10 and GATA2 at high levels. This ultimately suggests the anti-obesity effects of RFE by inhibiting adipogenic differentiation and lipid accumulation.

Adipogenic differentiation can be controlled by cell cycle regulation in the early phase of differentiation prior to the induction of transcription factor expression. Cell cycle regulation includes growth arrest, cell cycle reentry through clonal expansion, and subsequent growth arrest prior to terminal differentiation [28]. In the early stage of adipogenic differentiation, preadipocytes in a cell cycle-arrested state reenter the cell cycle through MCE upon receiving differentiation-inducing signals. Subsequently, another growth arrest occurs, leading to terminal adipogenic differentiation [18]. Thus, postponing cell cycle progression during early differentiation may be an effective strategy for inhibiting adipogenesis. In this study, we found that RFE delays the initiation of MCE, maintaining a high proportion of cells in the G0/G1 phase even after differentiation stimulation during the early phase of differentiation. Through this, we propose another mechanism of adipogenesis inhibition for RFE: postponement of the initiation of MCE, an essential step in early adipocyte formation.

AMPK is a promising therapeutic target for obesity, insulin resistance, and type 2 diabetes. It governs energy balance in response to metabolic stress and is an essential regulator of lipid and glucose homeostasis, as well as insulin sensitivity, in insulin-target tissues. In particular, the activation of AMPK, which is associated with lipid metabolism, can effectively inhibit adipogenic differentiation and lipogenesis by suppressing the expression of key transcription factors such as PPARγ and C/EBPα, as well as lipogenesis-related factors like SREBP-1, FAS, and ACC [29]. Also, AMPK can promote intracellular glucose uptake in adipose tissue by inducing the expression of the glucose transporter GLUT4, independently of the insulin signaling pathway [17]. As an insulin-responsive glucose transporter found in adipose and muscle tissues, GLUT4 expression levels can be utilized to evaluate intracellular glucose uptake [30]. Therefore, the activation of AMPK within adipose tissue may be an effective strategy for improving overall lipid metabolism and insulin sensitivity. We confirmed that RFE induces the activation of AMPK in adipocytes and adipose tissue while simultaneously enhancing intracellular glucose uptake through enhanced basal GLUT4 expression. These results indicate that the AMPK pathway may be involved in RFE’s anti-obesity effect and its ability to ameliorate insulin sensitivity.

Black fruits of plants in the genus *Rubia* are a potential source of anthocyanins and natural flavonoids [31,32]. Specific flavonoids that are known to be present in the genus *Rubia* include rutin, quercetin, kaempferol, rhamnetin, and apigenin derivatives [14,33]. We confirmed that rutin exists in RFE through LC-MS and HPLC analyses. It is known that rutin inhibits adipogenic differentiation and lipogenesis via the activation of AMPK in 3T3-L1 adipocytes and improves glucose uptake through the regulation of GLUT4 expression [23]. Therefore, we propose rutin as a bioactive component based on the LC-MS and HPLC chromatograms from this study, along with the results regarding the effects of RFE on obesity and insulin sensitivity. However, since RFE is an extract containing a variety of compounds, it cannot be assumed that the bioactivity of RFE results solely from rutin, and effects and interactions with and among other compounds are possible. In this regard, we plan to conduct follow-up studies comparing the bioactivity and rutin content of RFE with other natural products containing rutin. In conclusion, this study demonstrated the anti-obesity and insulin sensitivity-ameliorating effects of RFE and proposes that it is a natural product-derived material that could improve not only obesity but also its associated complications.

## 4. Materials and Methods

### 4.1. Preparation of RFE

The plant extract (KPM026-017) was procured from the Natural Product Central Bank at the Korea Research Institute of Bioscience and Biotechnology (Cheongju, Republic of Korea). It was gathered from Samjang-myeon, Sancheong-gun, Gyeongsangnam-do, Republic of Korea, in 2004. Dried fruit material (34 g) was ground, and 1 L of HPLC-grade methanol was added. Extraction was then conducted using an ultrasonic extractor (SD-Ultrasonic Co., Seoul, Republic of Korea), which involved 30 cycles of 15 min of sonication (40 kHz, 1500 W), each followed by a standing period of 120 min. Afterward, 5.97 g of RFE was obtained through filtration and low-pressure drying. At a concentration of 50 mg/mL, RFE was dissolved in dimethyl sulfoxide (DMSO) and subsequently diluted for application.

### 4.2. Cell Culture and Differentiation

Mouse 3T3-L1 preadipocytes were acquired from ATCC (Manassas, VA, USA) and maintained in Dulbecco’s modified Eagle’s medium with high glucose (DMEM-H) consisting of 10% new bovine calf serum (NBCS) and 1% (*v*/*v*) penicillin–streptomycin (Gibco, Paisley, UK). To induce differentiation into mature adipocytes, preadipocytes were plated in a 6-well plate (1 × 10^5^ cells/well). Once the cells reached 100% confluence, the post-confluent state essential for differentiation, they were incubated for an additional 2 days. Then, differentiation was induced by replacing the differentiation medium containing methylisobutylxanthine (0.5 mM), dexamethasone (0.25 mM), insulin (1 μg/mL) and indomethacin (0.125 nM) (MDI; Sigma, Saint Louis, MI, USA), and 10% fetal bovine serum (FBS; Gibco). On day 2 of differentiation, the medium was exchanged with DMEM-H consisting of 10% FBS and 1 μg/mL of insulin and then replaced every two days. On day 6 of differentiation, the medium was exchanged with DMEM-H supplemented only with 10% FBS. The adipogenic differentiation process was conducted for a total of 8 days.

### 4.3. Animal Experiments

Male C57BL/6 mice (six weeks old) were procured from HyoChang Science (Daegu, Republic of Korea). Before the experiment, all mice were acclimated to the controlled environmental conditions (12 h light/dark cycle, 25–30 °C) for 2 weeks. After this period, the mice were randomly classified into four groups, with six mice per group: (1) normal diet (ND) group, (2) high-fat diet (HFD) group, (3) group supplied RFE 12.5 mg/kg/day with HFD, and (4) group supplied RFE 25 mg/kg/day with HFD. The HFD consisted of 60% fat (5.24 kcal/g) and was obtained from Hyochang Science. The RFE group received RFE dissolved in phosphate-buffered saline (PBS) orally every two days, while the ND and HFD groups received the same amount of PBS orally. The body weight of each mouse was recorded every two days for 12 weeks, and food intake was recorded once a week. At 12 weeks, organs and blood samples were gathered for subsequent analyses. The animal experiments were endorsed by the Animal Ethics Committee of Kyungpook National University (Daegu, Republic of Korea) and were executed in concert with the institution’s ethical guidelines (approval number: KNU 2024-0244).

### 4.4. Cell Viability Assays

An MTT assay was administered to consider the cytotoxicity of RFE. First, 3T3-L1 preadipocytes (3 × 10^4^ cells/well) were seeded and maintained for 24 h. Once the cells reached 80% confluence, RFE was administered to the cells at doses of 6.25–100 µg/mL simultaneously with 0.0091% (*v*/*v*) DMSO as a control for 24 h. After the supernatant was discarded, the cells were incubated in 120 μL of an MTT solution for 3 h, and then 100 μL of isopropyl alcohol was added. After an hour, a 96-well plate reader (Tecan, Männedorf, Switzerland) was used to determine the absorbance of each well at 595 nm.

### 4.5. Oil Red O Staining

On 6 days of differentiation, maturely differentiated adipocytes were washed with PBS and fixed with 4% paraformaldehyde (PFA; Biosesang Inc., Yongin-si, Republic of Korea) for 1 h. After that, oil red O solution (Sigma) was applied to dye lipid droplets for 30 min. Then, the samples were washed with distilled water (dH_2_O) to discard excess dye, and the remaining moisture in the stained lipid droplets was allowed to dry. For quantification, isopropyl alcohol was added. The absorbance of dyes dissolved in the isopropyl alcohol was gauged at a wavelength of 450 nm.

### 4.6. Real-Time Reverse-Transcription Polymerase Chain Reaction (RT-PCR)

RNAiso Plus Reagent (TRIzol; TaKaRa Bio, Kyoto, Japan) was used to isolate RNA from fully differentiated adipocytes, and complementary DNA synthesis was accomplished with a ReverTra Ace™ qPCR RT Master Mix kit (TOYOBO, Kyoto, Japan). For the RT-PCR, an iCycler iQ™ real-time PCR detection system (Bio-Rad Laboratories, Hercules, CA, USA) was employed with an SYBR green master mix (Toyobo) and specific primers (Macrogen, Seoul, Republic of Korea; Appendix A). The mRNA expression levels of each factor were standardized using β-actin as a reference.

### 4.7. Western Blot Analysis

Proteins were isolated from fully differentiated adipocytes and adipose tissue with RIPA buffer (Biosesang Inc.) and PRO-PREP™ (Intron Biotechnology, Seoul, Republic of Korea). The Bradford assay (Bio-Rad laboratories) was employed for protein quantification, using bovine serum albumin as a standard. Protein lysates (25–30 μg of protein) were combined with 5X SDS-loading buffer (Biosesang Inc.) and heated at 100 °C for 10 min. To separate proteins by size, they were loaded onto SDS–polyacrylamide gels and electrophoresis performed. A nitrocellulose membrane was used to receive separated proteins. To eliminate nonspecific protein binding, 5% skim milk was used to block the transferred membranes for 1 h. Afterward, each membrane was maintained with specific primary antibodies (Appendix A) at 4 °C. Next day, the membranes were subjected to three washes with TBS-T buffer and subsequently treated with secondary antibodies linked to horseradish peroxidase (HRP) for 1 h. After washing again, the protein bands were detected by an enhanced chemiluminescence ECL detection kit (GE Healthcare, Buckinghamshire, UK) and a Fusion Solo detector. The expression levels of each protein were computed using ImageJ software version 1.53t and standardized to β-actin.

### 4.8. Flow Cytometry

To study RFE-induced cell cycle arrest at the early phase of differentiation, adipocytes underwent treatment with MDI (differentiation inducers) and RFE for 16 and 24 h, respectively. The adipocytes were then detached by exposure to a trypsin–EDTA solution (TE; Thermo Fisher Scientific, Waltham, MA, USA). After washing with 1X PBS containing 2% FBS, individual cells were treated with chilled 95% ethanol at 4 °C. The ethanol with cells was separated by centrifugation at 3000 rpm for 5 min. For staining, 100 μg/mL RNase A and 100 μg/mL propidium iodide were sequentially appended to the cell pellets and incubated for 30 min each. Cell cycle arrest analysis was conducted using an Attune acoustic focusing cytometer (Thermo Fisher Scientific).

### 4.9. Blood Biochemical Analysis

After euthanizing the mice, blood was harvested from the intra-orbital vein of the mice. To separate the plasma for analysis, samples of blood were spun for 20 min at 3500 rpm. The plasma levels of CHO, HDL-C, LDL-C, TG, glutamic oxaloacetic transaminase (GOT), glutamic pyruvic transaminase (GPT), blood urea nitrogen (BUN), and creatinine (CREA) were ascertained using an Olympus AU400 analyzer (Olympus Optical, Tokyo, Japan).

### 4.10. Hematoxylin and Eosin Staining

Inguinal and epididymal white adipose tissues (iWAT and eWAT, respectively) were fixed in 4% paraformaldehyde overnight to retain their integrity. The fixed tissues were paraffin-implanted and sectioned to a thickness of 5 μm. Then, each section was dyed using hematoxylin and eosin (H&E). Stained images of adipose tissue were obtained with a microscope, and the area (μm^2^) of lipid droplets within the adipose tissue was quantified using ImageJ software version 1.53t for four mice per group.

### 4.11. Intraperitoneal Glucose Tolerance Tests (IP-GTTs)

IP-GTTs were conducted during the 11 weeks of the experiment. The mice were deprived of food for 12 h before receiving an injection of 1 g/kg D-glucose. Blood glucose levels were checked from the tail vein using an Accu-Chek EZ glucose monitor (Roche Molecular Biochemicals, IN, USA) 0, 15, 30, 60, 90, and 120 min after administration.

### 4.12. Enzyme-Linked Immunosorbent Assay (ELISA)

Plasma was separated from blood samples obtained via the intra-orbital vein at 3500 rpm for 20 min and subsequently analyzed. Plasma insulin levels were ascertained using a mouse insulin ELISA kit (ALPCO, Salem, MA, USA) based on the manufacturer’s instructions. Plasma samples were analyzed in triplicate. To ascertain insulin resistance, the homeostatic model assessment for insulin resistance (HOMA-IR) was utilized, and it was estimated using the formula: HOMA-IR = (glucose [mg/dL] × insulin [μU/mL])/405.

### 4.13. LC-MS Analysis

Total ion chromatograms for RFE were obtained using an Alliance e2695 HPLC Separations Module combined with a Waters 2489 UV/vis detector and a QDa detector (Waters Corp, Milford, CT, USA). For chromatographic analysis, water (solvent A) and acetonitrile (solvent B), each containing 0.1% formic acid, were passed through an Acquity UPLC C18 column (250 mm × 3.0 mm, 5 µm) at 1 mL/min. After maintaining baseline equilibration for 10 min at 90% A, the elution conditions were as follows: 90–50% A for 0–20 min, 50–0% A for 20–30 min, 0% A for 30–35 min, 0–90% A for 35–35.01 min, and 90% A for 35.01–40 min to re-equilibration. For the analysis, 1000 ppm of RFE diluted in methanol was injected into 30 µL samples. Positive electrospray ionization mode was employed for mass spectrometry, scanning between 100 and 800 m/z, with the capillary voltage set to 0.8 kV. At a probe temperature of 400 °C and a source temperature of 120 °C, the collision energy increased to 15–40 V.

### 4.14. HPLC Analysis

An HPLC system (Waters 2695; Waters Corp.) coupled with a photodiode array (PDA) detector (Waters 2996; Waters Corp.) was utilized for the HPLC analysis. The wavelength range for detecting RFE and rutin was established as 210 to 400 nm. The stationary phase consisted of a C18 column (250 mm × 3.0 mm, 5 μm), while acetonitrile (solvent A) and 0.1% trifluoroacetic acid in water (solvent B) made up the mobile phase, flowing at a rate of 0.8 mL/min. After maintaining baseline equilibration for 10 min at 90% B, the elution conditions were as follows: 90–50% B for 0–20 min, 50–0% B for 20–30 min, 0% B for 30–35 min, 0–90% B for 35–35.01 min, and 90% B for 35.01–40 min to re-equilibration. For the analysis, 1000 ppm of RFE diluted in methanol was injected at 30 µL, and 200 ppm of rutin diluted in methanol was injected at 10 µL.

### 4.15. Statistical Analysis

Three biologically and technically independent repetitions were undertaken for all experiments, and the data are presented as means ± standard deviation (SD; for in vitro experiments) or means ± standard error of the mean (SEM; for in vivo experiments). Statistical analyses were appraised using one-way ANOVA conducted with GraphPad Prism 9.4.1 software (San Diego, CA, USA). A *p*-value below 0.05 was deemed to indicate statistical significance.

## Figures and Tables

**Figure 1 ijms-26-01833-f001:**
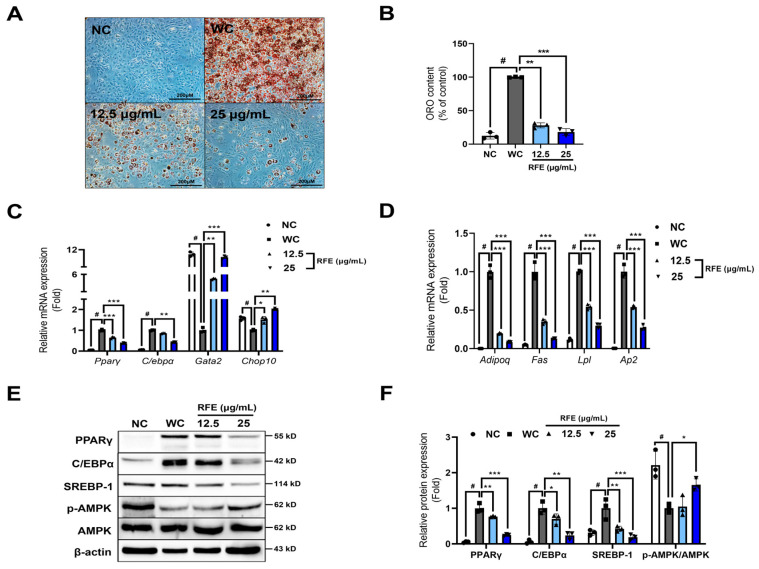
Effects of *Rubia akane* Nakai fruit extract (RFE) on adipogenic differentiation and lipogenesis in 3T3-L1 adipocytes. The in vitro experiment was conducted by dividing samples into four groups based on the presence or absence of methylisobutylxanthine, dexamethasone, insulin, indomethacin (MDI; differentiation inducers), and RFE treatment. (**A**,**B**) On day 6 of differentiation, lipid droplets were detected using oil red O staining, and then images were obtained ((**B**); scale bar: 200 μM) and the absorbance of dyes was gauged at a wavelength of 450 nm. (**C**,**D**) On day 8 of differentiation, the mRNA expression levels of adipocyte differentiation-related transcription factors and lipid accumulation markers in mature adipocytes were measured by real-time reverse-transcription polymerase chain reaction (RT-PCR). (**E**,**F**) On day 8 of differentiation, the protein expression levels of PPARγ, C/EBPα, SREBP-1, and AMPK in mature adipocytes were appraised by Western blot analysis. Three biologically and technically independent repetitions were undertaken for all experiments, and the results are depicted as the mean ± standard deviation (SD). * *p* < 0.05, ** *p* < 0.01, *** *p* < 0.001 compared with the WC (control; differentiated 3T3-L1 cells) group and DMSO group, ^#^ *p* < 0.01 when compared with the NC (non-differentiated 3T3-L1 cells) group.

**Figure 2 ijms-26-01833-f002:**
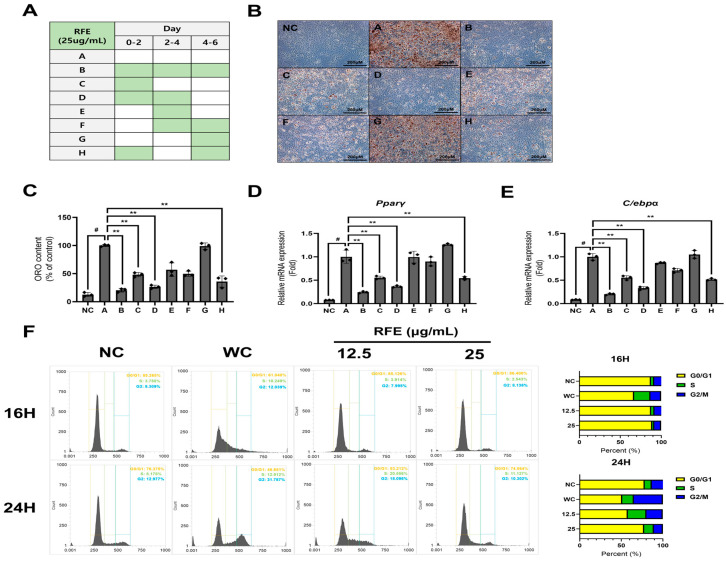
Effects of RFE in early phase of differentiation in 3T3-L1 adipocytes. To investigate the effect of RFE in each differentiation phase, RFE was applied at different intervals every two days during a total of six days of differentiation. (**A**) Diagram showing RFE treatment during adipocyte differentiation phases. (**B**,**C**) On day 6 of differentiation, lipid droplets were detected by ORO staining. Images were then obtained (B; scale bar: 200 μM), and ORO content was computed by determining the absorbance at 450 nm. (**D**,**E**) On day 8 of differentiation, the mRNA expression levels of *Pparγ* and *C/ebpα* in mature adipocytes were measured using RT-PCR, with expression levels normalized to β-actin. (**F**) Cell cycle regulation by RFE was assessed using flow cytometry after 16 and 24 h of treatment. Three biologically and technically independent repetitions were undertaken for all experiments, and the results are depicted as the mean ± SD. ** *p* < 0.01 compared with the A group, ^#^ *p* < 0.01 when compared with the NC group.

**Figure 3 ijms-26-01833-f003:**
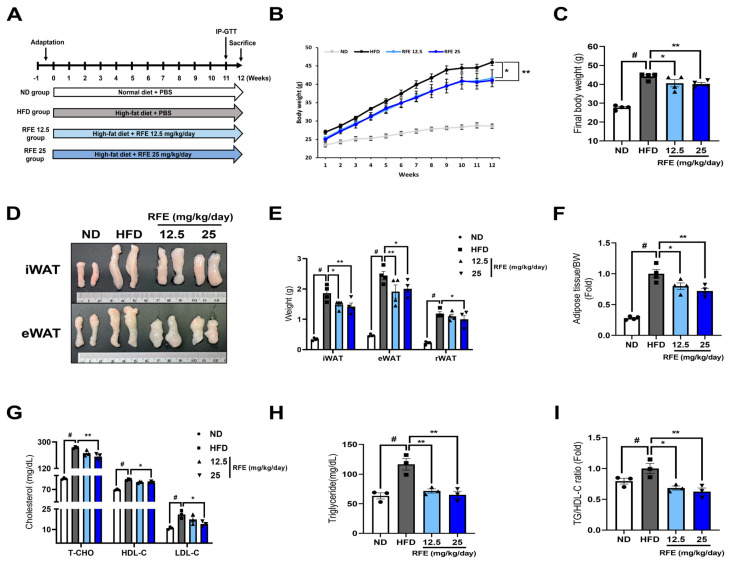
Effects of RFE on high-fat diet (HFD)-induced obesity and dyslipidemia in obese mice. In vivo experiments were conducted in four groups: the normal diet (ND) and HFD group were fed with PBS, and RFE group mice received either 12.5 mg/kg/day or 25 mg/kg/day of RFE along with an HFD. (**A**) Schematic diagram of the in vivo experiment. (**B**) Body weight was measured every other day for a total of 12 weeks. (**C**) At 12 weeks, the body weight of each group was measured (n = 4 mice/group). (**D**) Images and morphological changes of white adipose tissue (WAT), including inguinal WAT (iWAT) and epididymal WAT (eWAT). (**E**) At 12 weeks, each type of white adipose tissue, iWAT, eWAT, and retroperitoneal WAT (rWAT), was gathered and weighed. (**F**) Adiposity was determined by dividing the adipose tissue weight by the total body weight and expressed as fold changes relative to the HFD group (n = 4 mice/group). (**G**,**H**) At the end of the experiment, blood was gathered from each mouse to measure plasma CHO and TG levels. (**I**) The TG/HDL-C ratio is the plasma triglyceride content divided by the HDL-C content, expressed here as fold changes relative to the HFD group (n = 3 mice/group). Three biologically and technically independent repetitions were undertaken for all experiments, and the results are depicted as the mean ± standard error of the mean (SEM). * *p* < 0.05, ** *p* < 0.01, compared with the HFD (control) group, ^#^ *p* < 0.01 when compared with the ND group.

**Figure 4 ijms-26-01833-f004:**
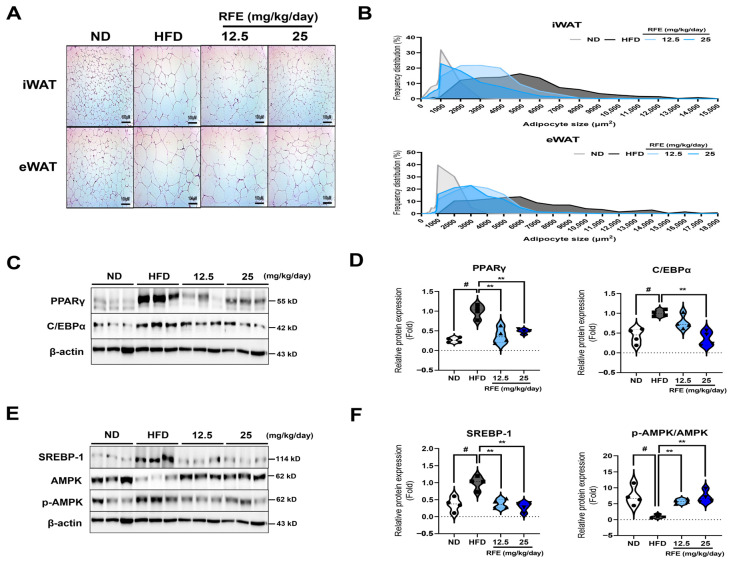
Effects of RFE on adipogenic differentiation and lipogenesis in white adipose tissue. After 12 weeks, white adipose tissues were isolated for analysis. (**A**,**B**) Microscopic images of iWAT and eWAT stained with hematoxylin and eosin (H&E) were obtained (A; scale bar: 100 μM), and ImageJ software was used to quantify the area of lipid droplets in each tissue type. (**C–F**) The protein expression levels of PPARγ, C/EBPα, SREBP-1, and AMPK in iWAT were evaluated by Western blot analysis (n = 4 mice/group). Three biologically and technically independent repetitions were undertaken for all experiments s, and the results are depicted as the mean ± SEM. ** *p* < 0.01 compared with the HFD group, ^#^ *p* < 0.01 when compared with the ND group.

**Figure 5 ijms-26-01833-f005:**
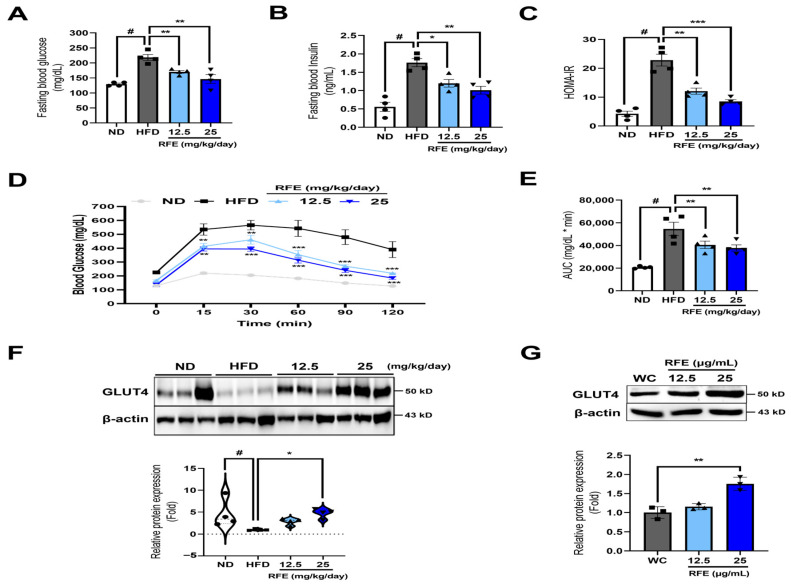
Effects of RFE on systematic insulin sensitivity in obese mice and 3T3-L1 adipocytes (n = 4 mice/group). (**A**,**B**) At 11 weeks, blood glucose and insulin levels were determined in mice deprived of food for 12 h utilizing an Accu-Chek EZ glucose monitor and an ELISA kit, respectively. (**C**) Insulin resistance was also measured using the homeostatic model assessment for insulin resistance (HOMA-IR; HOMA-IR = [glucose {mg/dL} insulin {μU/mL}]/405). (**D**,**E**) At 11 weeks, glucose (1 g/kg) was injected intraperitoneally into mice deprived of food for 12 h. Blood glucose levels were then measured 0, 15, 30, 60, 90, and 120 min after the injection through the tail vein, and the area under the curve (AUC) was computed and graphically represented. (**F**,**G**) The protein expression levels of GLUT4 in the eWAT of mice and in 3T3-L1 adipocytes were evaluated using Western blot analysis. Three biologically and technically independent repetitions were undertaken for all experiments, and the results are depicted as the mean ± SEM. * *p* < 0.05, ** *p* < 0.01, *** *p* < 0.001 compared with the HFD and WC group, ^#^ *p* < 0.01 when compared with the ND group.

**Figure 6 ijms-26-01833-f006:**
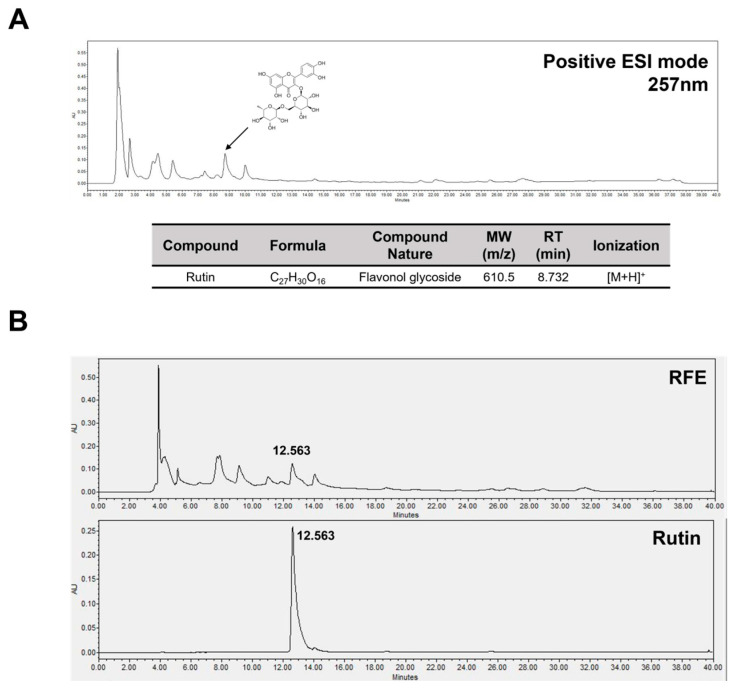
Liquid chromatography–mass spectrometry (LC-MS) and high-performance liquid chromatography (HPLC) analysis of RFE. Both LC-MS and HPLC profiling were conducted to analyze the bioactive compounds in RFE. (**A**) A total ion chromatogram of RFE produced by LC-MS analysis at UV 257 nm. (**B**) HPLC chromatograms for RFE (1 mg/mL) and rutin (0.2 mg/mL).

## Data Availability

Data are contained within the article and Appendix A.

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
