# Peer review of "Rubia akane Nakai Fruit Extract Improves Obesity and Insulin Sensitivity in 3T3-L1 Adipocytes and High-Fat Diet-Induced Obese Mice"

_ijms, 2025, doi:10.3390/ijms26051833_

Round 1
Reviewer 1 Report
Comments and Suggestions for Authors
The study investigates the potential anti-obesity effects of Rubia akane fruit extract (RFE). RFE was shown to inhibit adipogenesis and lipogenesis in 3T3-L1 adipocytes by activating AMPK and regulating key adipogenic genes, while also improving obesity and insulin resistance in high-fat diet-induced obese mice. These findings suggest that RFE could be a promising natural treatment for obesity and insulin resistance. The presented results are interesting, and the study was properly planned; however, I have a few comments.
Introduction:
· The introduction requires additional information about this plant, including the potential bioactive compounds it may contain.
· “Anti-obesity medications in Korea are categorized as appetite suppressants, fat absorption inhibitors, and glucagon-like peptide-1 (GLP-1) analogs”. A similar classification exists worldwide, not just in Korea, so I suggest refraining from mentioning any specific country.
Results:
· “In this paragraph, we assessed the inhibitory effects of RFE on adipogenic differentiation and lipogenesis in 3T3-L1 adipocytes”. The effects were assessed in the experiment, not in the paragraph. A similar mistake appears in several places throughout the manuscript.
· The graphs are too small and therefore unreadable. In Fig1B and Fig2B, nothing is visible. Perhaps these images could be placed in the supplementary section with better resolution and larger size?
· In Fig. 4, SD should be used instead of SEM. Have the assumptions for using the One-Way ANOVA test been checked? The authors should familiarize themselves with the proper guidelines for reporting statistical results. See: 10.4103/2141-9248.117957 or 10.1124/mol.119.118927
Methods:
· What was the final concentration of DMSO in the vehicle group in the cell experiment?
· By what route were the extracts administered to the mice?
· What was the purpose of performing the LCMS analysis, since the only information obtained relates to rutin, which was also detected using HPLC-PDA?
· The authors should report the concentration of rutin in the prepared extract, given that they conducted chromatographic analysis.
Comments on the Quality of English Language
There are spelling and grammatical errors in several places, and some sentences are not entirely clear.
Reviewer 2 Report
Comments and Suggestions for Authors
Overview
This paper attempts to study the mechanism of RFE in improving metabolic obesity and insulin resistance by regulating the expression of some fat-related factors in 3T3-L1 through cell and animal experiments. However, the experimental design of the study was too simplistic, and the causal relationship of the change could not be determined without rescue experiments, and more seriously, the statistical strategy of the study made major problems with the authenticity of the data in the article, and the authors' team seemed to be unable to distinguish which group was the real control group, so the reliability of the data was not high. We believe that the current research data are insufficient for publication.
Details
1. For the sake of data transparency, all bar charts should be changed to bar chart plus scatter chart format.
2. Protein and gene expression does not conform to a normal distribution, so data cannot be expressed using means and standard deviations, nor can ANOVA be used for normal distribution. The current method of statistical analysis is wrong and should be replaced by a nonparametric test.
3. The resolution of the figures is too small to see each data clearly, so the author is asked to provide a clearer version to reformat.
4. In addition to fat weight and tissue morphology, adipose is also associated with thermogenesis, and authors should supplement the results of thermogenesis-related tests.
5. Image issue: All western blots should be labeled with molecular weight.
6. The statistical difference annotation in all histograms is wrong, it is impossible for the NC or ND group to annotate the difference, and the Figure 1CDEG, 2CDE,3CDFGHIJ,4F,5ABCDEF has a similar error, please re-confirm the statistics.
7. The author's proprietary terminology expression is not standardized, for example, HDL and LDL, which should be expressed as HDL-C and LDL-C. There are many similar issues, and the author should revise the entire text.
8. There was a major problem with the data analysis of the study, and the average of the relative expression levels after fold change was not correct in the control group, which seriously affected the judgment of the results. We have doubts about the authors' team's data processing capabilities and data authenticity.
9. The biggest problem of this paper is that only simple cell experiments were conducted to observe biofactor changes under single treatment, no pathway-related phenotypic rescue experiments were set up, and no in vivo experiments were performed to verify the results, so the existing evidence is not conclusive.
10. 3T3-L1 cells are more representative of adipogenic differentiation, and the authors do not appear to have studied them.
Comments on the Quality of English LanguageThe English could be improved to more clearly express the research.
Round 2
Reviewer 1 Report
Comments and Suggestions for Authors
The manuscript has been improved after the revisions, and thanks for adding the missing required information.